# Mitoxantrone and Mitoxantrone-Loaded Iron Oxide Nanoparticles Induce Cell Death in Human Pancreatic Ductal Adenocarcinoma Cell Spheroids

**DOI:** 10.3390/ma16072906

**Published:** 2023-04-06

**Authors:** Jonas Dinter, Ralf P. Friedrich, Hai Yang, Christian Pilarsky, Harald Mangge, Marina Pöttler, Christina Janko, Christoph Alexiou, Stefan Lyer

**Affiliations:** 1Department of Otorhinolaryngology, Head and Neck Surgery, Section of Experimental Oncology and Nanomedicine (SEON), Else Kröner-Fresenius-Stiftung Professorship, Universitätsklinikum Erlangen, 91054 Erlangen, Germany; 2Medical Faculty, Friedrich-Alexander-Universität Erlangen-Nürnberg (FAU), 91054 Erlangen, Germany; 3Department of Surgery, Universitätsklinikum Erlangen, 91054 Erlangen, Germany; 4Clinical Institute of Medical and Chemical Laboratory Diagnosis, Medical University of Graz, 8036 Graz, Austria; 5Department of Otorhinolaryngology, Head and Neck Surgery, Section of Experimental Oncology and Nanomedicine (SEON), Professorship for AI-Controlled Nanomaterials, Universitätsklinikum Erlangen, 91054 Erlangen, Germany

**Keywords:** pancreatic ductal adenocarcinoma, 3D cell culture, nanomedicine, superparamagnetic iron oxide nanoparticles, magnetic drug targeting

## Abstract

Pancreatic ductal adenocarcinoma is a hard-to-treat, deadly malignancy. Traditional treatments, such as surgery, radiation and chemotherapy, unfortunately are still not able to significantly improve long-term survival. Three-dimensional (3D) cell cultures might be a platform to study new drug types in a highly reproducible, resource-saving model within a relevant pathophysiological cellular microenvironment. We used a 3D culture of human pancreatic ductal adenocarcinoma cell lines to investigate a potential new treatment approach using superparamagnetic iron oxide nanoparticles (SPIONs) as a drug delivery system for mitoxantrone (MTO), a chemotherapeutic agent. We established a PaCa DD183 cell line and generated PANC-1^SMAD4 (−/−)^ cells by using the CRISPR-Cas9 system, differing in a prognostically relevant mutation in the TGF-β pathway. Afterwards, we formed spheroids using PaCa DD183, PANC-1 and PANC-1^SMAD4 (−/−)^ cells, and analyzed the uptake and cytotoxic effect of free MTO and MTO-loaded SPIONs by microscopy and flow cytometry. MTO and SPION–MTO-induced cell death in all tumor spheroids in a dose-dependent manner. Interestingly, spheroids with a SMAD4 mutation showed an increased uptake of MTO and SPION–MTO, while at the same time being more resistant to the cytotoxic effects of the chemotherapeutic agents. MTO-loaded SPIONs, with their ability for magnetic drug targeting, could be a future approach for treating pancreatic ductal adenocarcinomas.

## 1. Introduction

Pancreatic cancer is the seventh leading cause of death in the world, with an increasing incidence [1,2]. The most prevalent tumor type among pancreatic cancers is pancreatic ductal adenocarcinoma (PDAC), with an overall 5-years survival rate of 11%, and it is expected to become the second leading cause of cancer death by 2030 [3]. Its poor prognosis is strongly associated with the time point of diagnosis, especially when the carcinoma has already spread beyond the pancreas [4].

Recent genomic analyses identified 32 mutated genes in pancreatic adenocarcinoma, which are divided into four subgroups correlating with histopathological characteristics, namely immunogenic, pancreatic progenitor, squamous and aberrantly differentiated endocrine/exocrine tumors [5]. The genetic drivers of PDAC are predominantly mutations in the well-known cancer genes *KRAS*, *CDKN2A*, *TP53* and *SMAD4*, besides other genes mutated at a lower prevalence [6]. SMAD4 mutations are correlated with worse clinical outcomes and these tumors are more prone to metastasize at a higher incidence in metastatic recurrence [7,8]. SMAD4 is capable of enabling gene transcription and tumor suppression via the TGF-β signaling pathway. Thus, it can control tumor development by facilitating growth arrest and an induction of apoptosis.

The poor survival rate (5-year survival < 11%) is not only due to the lack of early detection and delayed presentation caused by non-specific symptoms, but also because of inadequate therapies. Currently, surgical resection is the only treatment that offers a potential cure for pancreatic cancer. Supportive therapy through the application of chemotherapeutic agents may improve survival rates. There is some evidence that promises further improvement in survival with the administration of chemoradiotherapy in the neo-adjuvant setting [2,9]. Future treatment of this refractory disease may involve immunotherapy utilizing checkpoint inhibitors, protein, or whole-tumor-cell vaccines and nanoparticle-based drugs [9,10].

Nanoparticles, in particular superparamagnetic iron oxide nanoparticles (SPIONs), have become a new agent for treatment and diagnosis in cancer [11,12,13,14]. SPIONs can not only be used for imaging and for diagnostics of diseases, but also for enhancement of the accumulation and release of drugs at the pathological site, thereby increasing therapeutic efficacy and reducing the occurrence of side effects by decreasing their localization in healthy tissues [15,16]. Due to their magnetic properties, they can be controlled by an external magnetic field, allowing targeted delivery of therapeutics. This procedure, known as magnetic drug targeting (MDT), is characterized by high efficiency and effectiveness, as well as reduced side effects [17]. There are already several studies on the use of SPIONs for the diagnosis of pancreatic cancer using multimodal imaging, such as magnetic resonance imaging (MRI) and magnetic particle imaging (MPI) [18,19]. In contrast, there is only a limited amount of literature on the therapy of pancreatic cancer using functionalized SPIONs. For instance, the effects of SPIONs coated with curcumin and multifunctionalized magnetic iron oxide nanoparticles containing an anti-CD47 antibody, as well as the chemotherapeutic agent, gemcitabine, have been the subjects of former research [20,21]. However, MTO, a cytostatic drug used for cancer therapy and multiple sclerosis treatment with a potency up to 20,000 times higher than that of gemcitabine, has shown promising preclinical results for the treatment of pancreatic cancer [22,23,24,25]. Nevertheless, MTO did not show a significant response rate in a phase II clinical study including patients with advanced pancreatic carcinoma, as one of the main therapeutic limitations were dose-dependent hematologic side effects [26]. Since MTO not only effectively disrupts DNA synthesis and DNA repair, but also induces immunogenic cell death, the multiple effects of MTO combined with the targeting capacity of SPIONs might be a future approach for the therapy of pancreatic ductal adenocarcinoma [27,28]. The still predominantly used in vitro method to develop new drugs or agents for cancer therapy is a two-dimensional (2D) cell culture. However, this does not reflect tumor biology adequately due to unlimited amounts of oxygen and nutrition, as well as unphysiological changes in cell morphology [29]. Experiments performed with 3D cell cultures provide more accurate data on tumor characteristics, cell-to-cell interactions, drug sensitivity and metabolic profiles, and are more similar to the complex microenvironment cells experience in vivo [29,30,31].

In this study, we generated 3D spheroids of three pancreatic ductal adenocarcinoma cell lines (PaCa DD183 from primary tumors, wild-type PANC-1 and PANC-1^SMAD4 (−/−)^) by seeding the trypsinized cells into an agarose-coated 96-well plate. Free MTO was tested in for its efficacy to reduce the viability of cells and growth of tumor spheroids depending on their SMAD4 mutation status. Furthermore, we investigated the efficacy of SPIONs loaded with MTO in comparison with its unbound counterpart. Interestingly, we demonstrated an increased uptake of MTO, as well as SPIONs loaded with MTO, in *SMAD4*-mutated spheroids. In contrast, *SMAD4*-mutated cells showed a significantly worse response rate to the effects of MTO and SPIONs bound MTO.

## 2. Materials and Methods

### 2.1. Materials

Iron (II) chloride tetrahydrate (FeCl_2_·4H_2_O), iron (III) chloride hexahydrate (FeCl_3_·6H_2_O) and Sartorius ultrafiltration tubes Vivaspin 20, PES with a molecular weight cut-off (MWCO) of 100 kDa, were purchased from VWR (Darmstadt, Germany). Mitoxantrone was purchased from (Teva Pharm, Ulm, Germany). Iron reference standards were purchased from Bernd Kraft GmbH (Duisburg, Germany). Agarose, HCl, eosin, sterile Rotilabo^®^ syringe filters with cellulose-mixed ester membranes and dialysis bags (Repligen, Spectra/Por 6, MWCO 10 kDa) were supplied by Roth (Karlsruhe, Germany). Ringer’s solution was purchased from Fresenius Kabi AG, Bad Homburg, Germany. Propidium iodide (PI), Triton X-100, ribonuclease A, Dulbecco’s phosphate-buffered saline (PBS) and lauric acid were purchased from Sigma-Aldrich (St. Louis, MO, USA). Annexin A5-FITC (AxV-FITC), Hoechst 33342 and 1,1′,3,3,3′,3′-hexamethylindodicarbo-cyanine iodide (DiIC1(5)) were purchased from Life Technologies (Darmstadt, Germany). Cell culture plates were ordered from Sarstedt (Nümbrecht, Germany). Fetal bovine serum (FBS) was ordered from Thermo Fisher Scientific (Waltham, MA, USA) and trypsin/EDTA solution (0.05/0.02% in PBS) from Pan Biotech (Aidenbach, Germany). DMEM was purchased from Biochrom (Berlin, Germany), RPMI 1640 from Life Technologies (Carlsbad, CA, USA), and keratinocyte SFM from Gibco (Dublin, Ireland). CryoGlue embedding media were purchased from SLEE medical GmbH (Mainz, Germany), Hematoxylin Gill III and DPX mounting media from Merck KGaA (Darmstadt, Germany), and microscope slides from Menzel GmbH (Braunschweig, Germany). Water used in all experiments was of ultrapure quality produced by the Milli-Q^®^ system from Merck (Darmstadt, Germany).

### 2.2. Synthesis of Superparamagnetic Iron Oxide Nanoparticles (SPIONs)

Lauric acid (LA)-coated SPIONs were synthesized according to the protocol of Zaloga et al., 2014 [32]. Briefly, Fe (II) chloride and Fe (III) chloride were dissolved in ultrapure water and co-precipitated by stirring at 80 °C in alkaline media, under an argon atmosphere. After coating with lauric acid, the suspension was homogenized for 30 min at 90 °C and dialyzed several times (MWCO of 10 kDa) with ultrapure water. Subsequent coating with a protein corona of human serum albumin (HSA) was performed according to the protocol of Zaloga et al., 2016 [33]. Briefly, HSA solution (Recombumin Elite, 10% *w/v*, Albumedix, Nottingham, UK) was sterilized using a 0.22 µm filter, transferred into dialysis bags (MWCO 10 kDa) and dialyzed against 4.5 L of ultrapure water (4 water changes, 5 h). After concentration by tangential flow ultrafiltration [34], the HSA solution was stirred at room temperature and lauric-acid-coated SPIONs were added dropwise through a 0.8 µm filter. Finally, excess HSA was removed by tangential flow ultrafiltration before the SPION solutions were filtered using a 0.22 µm sterile filter. MTO loading of SPIONs was performed as reported by Zaloga et al., 2016 [33]. Briefly, 900 µL of SPIONs (4.84 mg Fe/mL) was vortexed with 100 µL mitoxantrone solution (2 mg/mL) and incubated for 5 min, to obtain a SPION^MTO^ solution containing 200 µg/mL MTO and 4.36 mg/mL iron. Further characterization of MTO binding to SPIONs and analysis of binding stability have been previously performed by our work group, reported by Zaloga et al., 2016 [33].

### 2.3. Iron Quantification of Nanoparticles

The iron content of the SPIONs was determined using an Agilent 4200 microwave plasma-atomic emission spectrometer (MP-AES) (Agilent Technologies, Santa Clara, CA, USA). The SPION suspension (50 µL) was dissolved in 50 µL 65% HNO_3_, and incubated for 10 min at 95 °C. Prior to analysis, the sample was further diluted in 1900 µL ultrapure water. A commercial iron solution was used as an external standard.

### 2.4. Dynamic Light Scattering (DLS) and Zeta Potential Measurements

A Malvern Zetasizer (Malvern Instruments, Worcestershire, UK) was used to determine the zeta potential and the hydrodynamic size of the particles in water. Particles were diluted to an absolute iron content of 50 µg/mL and measured in triplicate at 25 °C.

### 2.5. Cells and Culture Conditions

PANC-1, a human pancreatic carcinoma cell line of ductal origin, was purchased from ATCC (Manassas, VA, USA). PANC-1 cell lines were cultivated in DMEM with 9% fetal bovine serum and 0.9% L-Glutamin, PaCa DD183 in DMEM, 13% FBS and 22% keratinocyte-SFM, and PANC-1^SMAD4 (−/−)^ in RPMI 1640, 9% FCS and 0.9% L-Glutamine. All cell lines were cultivated under standard cell culture conditions at 37 °C and 5.0% CO_2_, and passaged two times a week using trypsin/EDTA, according to the manufacturer’s instructions. For all experiments, cells were grown to a confluence of 70–90% in 75 cm^2^ cell culture flasks.

### 2.6. Generation of Pancreatic Carcinoma Cell Lines, DD183 and Panc-1^SMAD4 (−/−)^

PaCa DD183 and PANC-1^SMAD4 (−/−)^ cell lines were produced in the Department of Surgery, Universitätsklinikum Erlangen, Germany, and are further described by Friedrich et al., 2020 [35]. PaCa DD183 cells were established from a pancreatic ductal adenocarcinoma using the Dresden outgrowth protocol, as described by Rückert et al. [36]. PANC-1^SMAD4 (−/−)^ cells were generated by using the CRISPR-Cas9 system on PANC-1 cells [37]. In short, pSpCas9(BB)-2A-GFP (PX458) (Addgene, #48138, RRID: Addgene_48138, Watertown, MA, USA) was used as a plasmid vector. sgRNAs were synthesized by Eurofins Genomics Germany GmbH. The sgRNAs were designed based on the human and mouse CRISPR knockout-pooled library (Addgene #1000000053, 1000000049) [38]. Plasmid construction was performed according to the protocol [37] and confirmed by sequencing. The cells were transfected with constructed plasmids by using Lipofectamine 3000 Transfection Reagent (Invitrogen, cat. #L3000015, Waltham, MA, USA) for 72 h. Afterward, the GFP-positive cells were selected by FACS sorting. Selected cells were seeded into two 96-well plates for selecting single clones. The knockout effect was detected by a Western blot.

The list of sgRNAs which were used in this study is a follows:

NC human sgRNA, forward: 5′-caccgATCGTATCATCAGCTAGCGC-3′

NC human sgRNA, reverse: 5′-aaacGCGCTAGCTGATGATACGATc-3′

smad4 human sgRNA1, forward: 5′-caccgAACTCTGTACAAAGACCGCG-3′

smad4 human sgRNA1, reverse: 5′-aaacCGCGGTCTTTGTACAGAGTTc-3′

smad4 human sgRNA2, forward: 5′-caccgAGTCCTACTTCCAGTCCAGG-3′

smad4 human sgRNA2, reverse: 5′-aaacAGTCCTACTTCCAGTCCAGGc-3′

### 2.7. Western Blotting

Cells were lysed with RIPA-buffer (Radioimmunoprecipitation assay buffer). Proteins were quantified with a BCA protein assay kit (Thermo Scientific, Catalog No. 23225, Waltham, MA, USA). Protein samples (10 µg) were separated by 10% SDS-PAGE and transferred onto a nitrocellulose membrane. The membrane was blocked with 5% non-fat milk solution at room temperature for one hour. Primary antibody SMAD4 (Santa Cruz Biotechnology Cat#sc-7966, RRID:AB_627905, Dallas, TX, USA) was used for the detection. GAPDH (Cell Signaling Technology Cat#5174, RRID:AB_10622025, Danvers, MA, USA) was used as the loading control. HRP (horseradish peroxidase)-linked anti-rabbit IgG (immunoglobulin G) (Cell Signaling Technology Cat#7074, RRID:AB_2099233) and HRP-linked anti-mouse IgG (Cell Signaling Technology Cat#7076, RRID:AB_330924) were used as the secondary antibodies. Quantification of signals was performed by Amersham Imager 600 (Pittsburgh, PA, USA), with the SignalFire Elite ECL Reagent (Cell Signaling Technology, 12757S).

### 2.8. Generation of Pancreatic Carcinoma Spheroids

Cellular spheroids were produced in 96-well plates coated with 50 µL of 1.5% agarose. Briefly, cells were washed with PBS and detached from the cell culture surface using trypsin/EDTA. After dilution in 10 mL cell culture medium, the number of viable cells was analyzed by an automatic cell counter (MUSE^®^ Cell Analyzer, Merck-Millipore, Billerica, MA, USA). Agarose-coated wells were equipped with 150 µL medium and subsequently seeded with 50 µL (300,000 viable cells/mL) cell suspension, and incubated for 72 h at 37 °C.

### 2.9. Determination of Spheroid Growth by Transmission Microscopy

On day three, five and seven after seeding, the spheroid size was determined by analyzing microscopy images using an Axiovert 40 CFL Microscope and the Axio Vision SE64 Rel4.9 software (Zeiss, Jena, Germany). The areas of the 2D projections of the spheroids were determined by the ImageJ software (National Institutes of Health, Bethesda, MD, USA). Experiments were conducted at least four times, with nine replicates each. Microsoft Excel was used for statistical analysis.

### 2.10. Hematoxylin/Eosin Staining of Spheroid Cryosections

Five representative spheroids from each cell line were selected for cryosection preparation. After removing the cell culture medium, the spheroids were carefully embedded in CryoGlue embedding medium. Twenty minutes later, spheroids were frozen at −20 °C for at least 24 h, before preparation of 10 µm sections using an MNT microtome (SLEE medical GmbH, Mainz, Germany). The sections were then transferred to slides until histological staining. For the staining, cryosections were washed with distilled water for 5 s before staining with Hematoxylin Gill III for 6 min. Subsequently, the sections were placed in 0.1% HCl for 5 s and rinsed with tap water for 3 min. Counterstaining was performed with 0.5% eosin for 6 min, followed by rinsing with tap water and embedding with DPX mounting medium. The sections were imaged and analyzed using an Axiovert 40 CFL Microscope and the Axio Vision SE64 Rel4.9 software (Zeiss, Jena, Germany).

### 2.11. Mitoxantrone Treatment of Spheroids

Seventy-two hours after seeding pancreatic cells in 96-well plates, spheroids were treated for an additional 4 days with various MTO concentrations (0, 0.5, 5, 10, 20 and 47.8 µg/mL) under standard cell culture conditions. Bright-field images were taken after three, five and seven days using an Axiovert 40 CFL Microscope and the Axio Vision SE64 Rel4.9 software (Zeiss, Jena, Germany). The areas of the 2D projections of the spheroids were determined by the ImageJ software (National Institutes of Health, Bethesda, MD, USA). Experiments were conducted four times, with technical triplicates. Microsoft Excel was used for statistical analysis.

### 2.12. Treatment of Pancreatic Spheroids with SPION, SPION^MTO^ and MTO

Pancreatic spheroids were treated with SPION (478 µg/mL Fe), SPION^MTO^ (20 µg/mL) and free MTO (20 µg/mL) in the same MTO and iron concentration, 72 h after seeding. Spheroids were incubated for an additional 96 h under standard cell culture conditions, followed by imaging using a Zeiss microscope (Axio Observer Z.1, Zeiss, Jena, Germany).

### 2.13. Dissolving of Spheroids to Single-Cell Suspensions

Spheroids were harvested from 96-well plates and pooled in 15 mL cell culture flasks (Falcon, Sarstedt, Nümbrecht, Germany). The supernatant was removed, and the spheroids were washed with 500 µL PBS for at least 1 min. Afterward, the spheroids were incubated with 150 µL trypsin for 20–30 min. Spheroids were dissolved by repeated pipetting. Cell culture medium (750 µL) was added to neutralize the trypsin, and the suspension was centrifuged at 300× *g* for 5 min before carefully removing the supernatant. The resulting cell pellet was finally resuspended in 250 µL of medium.

### 2.14. Analysis of Viability and Cell Death Phenotype in Flow Cytometry

To determine viability, 50 µL of the single-cell suspensions were incubated with 250 µL of the staining mixture for 20 min at 4 °C. One milliliter of the staining solution contained 10 μg Hoechst 33342, 25 ng AxV-FITC and 66.6 ng PI per mL Ringer’s solution. The fluorescence intensity was analyzed with a Gallios flow cytometer (Beckman Coulter, Fullerton, CA, USA). Excitation of FITC and PI was obtained at 488 nm; FITC fluorescence was verified with the FL1 sensor (525/38 nm band pass filter, BP); and the PI fluorescence with the FL3 sensor (620/30 nm BP). The MTO fluorescence was excited at 638 nm and recorded by the FL7 sensor (725/20 nm BP). Excitation of the Hoechst 33342 fluorescence was obtained at 405 nm and recorded by the FL9 sensor (430/40 nm BP). Electronic compensation was used to eliminate fluorescence bleed-through. Data analysis was conducted with the Kaluza software Version 1.2 (Beckman Coulter).

## 3. Results

### 3.1. Physicochemical SPION Characterization

The physicochemical properties greatly influence not only the interaction between nanoparticles and cells, but also the final quality of the loaded particles. Therefore, the quality and reproducibility of particle synthesis was controlled by measurements of particle size and zeta potential using dynamic light scattering (DLS) (Table 1). When freeze fracture transmission electron microscopy (TEM) was performed, we found that SPIONs formed multicore particles, of which the aggregate size was comparable to the size measured by dynamic light scattering (DLS). Every individual particle had a diameter of approximately 10 nm [33]. The zeta potential of SPIONs in ultrapure water showed a negative value of −21 mV. In water, the SPIONs exhibited hydrodynamic sizes of 65 nm. The absence of larger aggregates indicated high colloidal stability. In addition, previous studies demonstrated high colloidal stability of all particles in FCS-containing media, indicating the formation of a stabilizing protein corona and a successful binding of MTO to the particle surface [33,39]. In serum-containing medium and whole blood, even in the presence of a magnetic field, no permanent agglomerations were detected [40].

Thus, the physicochemical data suggest the suitability of the nanoparticles for subsequent in vitro experiments. MTO was freshly loaded onto the SPIONs before every experiment. Binding experiments indicated adsorption of 97.6 ± 0.1% of 500 µg MTO adsorbed to 1 mL of SPIONs (2 mg/mL) after 5 min equilibrium. The release of MTO from the particles incubated in RPMI-1640 medium was determined via dialysis and a magnetic assay setup. After 72 h, 11.6 ± 0.1% and 23.7 ± 0.4% were released from the former or the latter, respectively. Thus, MTO was released rather slowly from the particles, most likely by diffusion [33,39].

### 3.2. Generation of 3D Spheroids of Different PDAC Cell Lines

We used various PDAC cells lines to form spheroids. PaCa DD183 and PANC-1 originated from a primary PDAC [41]. Furthermore, the established cell line PANC-1 with intact SMAD4, as well with knocked-out SMAD4 (see Appendix A), was investigated. It is known that the SMAD4 gene is involved in gene regulation and tumor suppression via the TGF-β pathway. Furthermore, mutations are correlated with worse clinical outcomes and connected to higher metastasis rates [7,8].

After evaluating the optimal culture condition for each pancreatic ductal adenocarcinoma cell line, 15,000 cells were seeded for spheroid generation into agarose-coated 96-well-plates (Figure 1). After spheroid structures were formed, spheroid size was determined microscopically after 3, 5 and 7 days of incubation (Figure 1a). Analysis of transmission microscopy images revealed large cell-line-dependent size differences during the initial phase of spheroid formation, with PANC-1^SMAD4 (−/−)^ producing the smallest and PANC-1 the largest spheroids (Figure 1c). In addition, the images and image analysis showed that between the third and seventh day, the size of all spheroids decreased significantly, indicating an increase in spheroid density [42]. After seven days, the edges of all spheroids, except those formed by PaCa DD183 cells, finally began to fray. Cryosections prepared on the third, fifth, and seventh day and subsequently stained with hematoxylin/eosin (HE) to visualize necrotic areas, revealed no evidence of necrotic cores (Figure 1b). This is consistent with previous reports, which demonstrated that mass transport in small 3D cell aggregates below 400–500 µm is sufficient to maintain cell viability through adequate diffusion of nutrients, oxygen and metabolic wastes [43,44,45].

### 3.3. Impact of Free MTO on PDAC Spheroid Growth

The effect of MTO treatment on the growth and viability of PDAC cells spheroids was examined after the addition of free MTO (0, 0.5, 5, 10, 20 and 47.8 µg/mL) on the third day of spheroid formation and during the following four days (Figure 2). Bright-field images on days 3, 5 and 7 confirmed the size reduction of untreated spheroids, as shown in Figure 1.

Compared with the size of untreated spheroids at days 5 and 7, treatment with 20 µg/mL and 47.8 µg/mL MTO resulted in a significant concentration-dependent increase in the size of PaCa DD183 and PANC-1^SMAD4 (−/−)^ spheroids, indicating an effect on cell cohesion at higher MTO concentrations (Figure 2a,b,e,f). Interestingly, the size of PANC-1 spheroids increased sharply at already 5 µg/mL and 10 µg/mL MTO, whereas the increase was less pronounced after incubation with 20 µg/mL and 47.8 g/mL MTO, indicating a very rapid cell death at MTO concentrations of 20 µg/mL or higher (Figure 2c,d).

### 3.4. Impact of Free MTO on Cell Viability within PDAC Spheroids

At day seven, we analyzed the viability of the spheroids via flow cytometry (Figure 3a,b) to get a clearer insight into the mechanism of death induced after 96 h incubation with MTO. As they are indicators of apoptosis and necrosis, we evaluated the cells for their phosphatidylserine exposure using Annexin V-FITC (Ax), and for their plasma membrane integrity using propidium iodide (PI). Ax−PI are considered viable, Ax+PI apoptotic and PI+ cells necrotic.

The applied gating function, shown exemplarily for PaCa DD183 cells in Figure 3a, was used to analyze cell morphology, phosphatidylserine exposure and plasma membrane integrity and MTO uptake into the cells. PaCa DD183 cell spheroids revealed a dose-dependent increase in apoptotic and necrotic cells with increasing MTO concentrations. At the highest MTO concentration (47.8 µg/mL), the number of viable cells decreased to 20.1 ± 6.7% (Figure 3b). In contrast to PaCa-DD183, the sensitivity of PANC-1 cell spheroids against MTO was much more pronounced. A concentration of 5 µg/mL reduced the viability to 2.2 ± 0.7%, compared to 70.4 ± 6.0% in PaCa DD183 cells. Thereby, the percentage of apoptotic cells increased to 92.4 ± 1.3%, while PaCa DD183 cells only 17.2 ± 2.5% apoptotic cells in the presence of 5 µL/mL MTO. Interestingly, PANC-1 cells with a mutated SMAD4 gene were much less sensitive to MTO compared to the original PANC-1 cells. Thus, the use of 5 µg/mL MTO resulted in only 33.4 ± 14.8% apoptotic and 2.8 ± 2.3% necrotic cells, and only at the highest MTO concentration (47.8 µg/mL), the cells reached approximately the same values that were already reached in the PANC-1 cells at 5 µg/mL.

### 3.5. Impact of Particle-Bound MTO and Free MTO on PDAC Spheroids

Based on the cytotoxicity data obtained with free MTO on PDAC spheroids, 20 µg/mL was selected as the MTO concentration for the comparison of effects achieved in free or nanoparticle-loaded form. Spheroids treated with non-loaded SPIONs (478 µg/mL Fe) or H_2_O served as the controls. After dissociation of spheroids into single cells, analysis of viability was performed by flow cytometry (Figure 4a–c). In PaCa DD183 cell spheroids, treatment with free MTO and SPION-linked MTO resulted in similar cytotoxicity with comparable numbers of viable cells (34.7 ± 1.8% and 38.8 ± 10.2%, respectively). There were also no significant differences in PANC-1 viable cells between treatment with free and bound MTO, although the effect on PANC-1 cells was much stronger (8.9 ± 3.2% and 3.6 ± 1.8%, respectively).

## 4. Discussion

PANC-1^SMAD4 (−/−)^-derived spheroids similarly showed no significant differences in the number of viable cells after MTO or SPION^MTO^ treatment (23.3 ± 0.6% and 16.0 ± 1.0%, respectively). However, the proportion of viable cells is more pronounced than in PANC-1^SMAD4 (−/−)^ spheroids, confirming the decreased sensitivity to toxic substances in cells with downregulated or absent SMAD4 expression, as shown in Figure 3.

In PaCa DD183 and PANC-1 cells, we observed a decrease in the percentage of necrotic cells after SPION^MTO^-treatment compared to free MTO, which might be an indication for a slightly lower toxicity of SPION-bound MTO. However, this was not the case for PANC-1^SMAD4 (−/−)^ cells, which revealed slightly more necrosis when treated with SPION^MTO^ in comparison to the free drug.

To find an explanation for the cell-specific differences in MTO effects, the cellular MTO amount was determined by analyzing the fluorescence intensity (Figure 4d–f). The increased cytotoxicity of MTO toward PANC-1 cells compared with PaCa DD183 cells correlated with an increase in cellular MTO concentration. In PANC-1 cells, fluorescence intensity upon treatment with free MTO or SPION^MTO^ was 2.4- or 6.2-fold higher than in PaCa DD183 cells. Although the number of viable cells in PANC-1^SMAD4 (−/−)^ spheroids was higher than in spheroids derived from PANC-1 cells, the amount of MTO detected in PANC-1^SMAD4 (−/−)^ cells was 3.3-fold (free MTO) and 1.2-fold (SPION^MTO^) higher than in PANC-1 cells. This indicates a higher uptake of free and particle-bound drug into the SMAD4-deletion mutant cells.

Since all solid tumors only occur as 3D structures in vivo, 3D cell culture systems may provide a way to bridge the gap between 2D cell cultures and the in vivo setting. In particular, analyzing the effects of drugs in cells grown in a monolayer cannot reflect the complexity of the 3D structure of a tumor [29]. Usually, in vivo studies are performed using animal models to overcome these problems. However, as the enormous number of mice and other animals used for research should be drastically reduced, in vitro testing systems need to be adapted to better approximate the real situation, which could lead to a reduction in animal testing, according to the 3Rs principle (replace–reduce–refine).

Testing pancreatic ductal adenocarcinoma presents a particular challenge, as this tumor entity is represented by hypoxic conditions and a dense specialized extracellular matrix/desmoplastic stroma within these tumors. Two-dimensional cell culture systems cannot replicate these features. Moreover, it has been shown that culturing human pancreatic adenocarcinoma cells in monolayers leads to a transition from epithelial cells to mesenchymal phenotypes [46].

Although it has already been shown by others that PDAC cell lines are able to form 3D spheroids, our results demonstrate that we could consistently reproduce tumor spheroids from PaCa DD183 and PANC-1 cell lines, as well as the newly generated SMAD4 knock-out cell line, PANC-1^SMAD4 (−/−)^, to have a working platform for drug testing. Even though solid spheroids started to form after approximately 24 h, our goal was to obtain tightly packed tumor spheroids. Due to the 3D structure of the spheroid, the inner core has a limited amount of resources like oxygen and nutrients available, while metabolic waste needs to be transported out of the spheroid. Thus, the 3D model comes closer to reality than a 2D cell culture. One limitation of the used 3D spheroid model is the monoculture of PADC cells. In future studies this model can be further developed by coculturing cancer-associated fibroblasts and immune cells to improve resembling the characteristic dense stroma of PDACs [29,47].

After establishing highly reproducible tumor spheroids in PaCa DD183, PANC-1 and PANC-1^SMAD4 (−/−)^ tumor cell lines, we tested their resistance to mitoxantrone. In PaCa DD183 and PANC-1^SMAD4 (−/−)^ spheroids, MTO treatment resulted in a significant concentration-dependent increase in the area size (Figure 2a,b,e,f). Surprisingly, treatment of PANC-1 spheroids with MTO caused an increase in spheroid area at concentrations of 5 or 10 µg/mL MTO, while at lower and higher concentrations, the compact structure of the spheroids was not lost (Figure 2c,d). Whilst the non-dose-dependent behaviour is unusual at first, this might be explained by the fact that at concentrations of 5 and 10 µg/mL, apoptosis is massively induced in the spheroids, as detected by AxPI staining (Figure 4). Apoptosis is accompanied by the controlled shrinkage and condensation of the cell body and the degradation into multiple individual apoptotic bodies, which might explain the loss of coherence, and thus increase of area size. In contrast, MTO concentrations of 20 µg/mL might cause an immediate loss of plasma membrane integrity in PANC-1 spheroids due to an overload of toxic MTO, resulting in a higher percentage of necrotic cell death (Figure 3). As this process is very quick, there might be no time for controlled degradation, and therefore a reduced loss of coherence.

Furthermore, our results demonstrate how genetically diverse PDAC tumor cell spheroids can respond differently to a given chemotherapeutic agent. PaCa DD183 and PANC-1 originated from a primary PDAC [41]. In the PANC-1^SMAD4 (−/−)^ cell clone, SMAD4 expression was knocked out using CRISPR/Cas9. It was previously pointed out that PANC-1 tumor spheroids are more resistant to chemotherapy and radiotherapy when harboring *KRAS* and *p53* mutations compared to 2D cell cultures [48,49,50,51]. Therefore, we were interested in whether knocking out *SMAD4* in the PANC-1 cell line, which has been found to be associated with a worse prognosis in pancreatic carcinoma, would affect resistance to MTO [52,53,54]. Interestingly, the uptake of free MTO and SPION bound MTO was increased in PANC-1^SMAD4 (−/−)^ cells. These findings correspond with the increased uptake of SPIONs in PANC-1^SMAD4 (−/−)^ cells in comparison to PANC1 cells, demonstrated by our research group and reported by Friedrich et al. [35]. Nevertheless, they showed a higher rate of viability compared to wildtype SMAD4 cells. Even though increased chemoresistance has been shown for other chemotherapeutic substances [55], to our knowledge this is the first study demonstrating an increased resistance to MTO in PANC-1 spheroids due to SMAD4 knockout.

We chose MTO and MTO-loaded SPIONs in this study because of their high chemotherapeutical potency and capabilities of inducing immunogenic cell death by the release of damage-associated molecular patterns. At the same time, MTO is quite stable, making it excellent for analysis even in complex media [22,23,24,56]. Our data point out that MTO-loaded SPIONs, despite their slightly impaired penetration in comparison to unbound MTO, induce cell death reliably in pancreatic carcinoma spheroids. 

Further research is needed to differentiate if MTO is released from the particles or if the particles penetrate into the spheroid together with the chemotherapeutic cargo. In previous investigations, with spheroids from HT-29 colon carcinoma cells, we observed a delayed penetration of MTO into the tumor spheroids when applied as SPION–MTO, by using fluorescence microscopy [57]. We hypothesize that the MTO must have been bound to the SPIONs at least initially, otherwise we would not have seen differences in uptake velocity and amount.

The cytotoxic potential of MTO comes along with severe dose-dependent side effects, which are the main therapeutical limitation for use in patients [26]. As demonstrated previously by our research group, MDT using MTO-loaded SPIONs could be a way to increase intratumoral cytotoxicity, while at the same time reducing systemic side effects [16,58,59]. It has been shown earlier that using an immunogenic cell death inductor conjugated on nanocarriers can induce a significant tumor reduction in pancreatic carcinomas [60]. Our group demonstrated previously that the loading of MTO onto SPIONs does not influence the capability of the drug to induce immunogenic cell death in tumor cells [27]. We also showed previously that the chemotherapeutic drug MTO can be accumulated in the tumor region, and the immune system was spared from the toxic effects of the drug [16]. The conservation of the immune system and the simultaneous induction of immunogenic cell death in the tumor region might improve therapeutic efficacy in PDAC.

Another possible future therapeutic application is the ability of SPIONs to locally induce hyperthermia by using alternating magnetic field-induced movement [61]. This could lead to an increased anti-tumoral activity, as it has been shown that mild hyperthermia can increase the cytotoxic effects of MTO [62,63].

With further research, combining the previously described features of SPIONs and MTO could open up possibilities for new, more effective treatment approaches regarding pancreatic carcinomas.

## 5. Conclusions

Our study demonstrated that 3D PDAC cell spheroids are a reproducible tool for in vitro assays. We found that nanoparticle-loaded MTO induced cell death in 3D PDAC cell spheroids. Furthermore, knocking out *SMAD4* in PANC-1 cell spheroids led to an increased uptake of unbound and bound MTO. Interestingly, we demonstrated an increased cell resistance to the cytotoxic effects of free and bound MTO. This highlights the importance of the *SMAD4* mutation for future treatment attempts. Potentially, a deeper understanding of genetic key mutations such as SMAD4 could enable an individualized treatment with an improved clinical outcome. Further research is needed to evaluate SPION potential for MDT in the field of pancreatic carcinomas. Locally increasing the intratumoral drug concentration, as well as reducing systemic side effects via MDT, could be a future approach to enhance the therapeutic options for patients with pancreatic carcinomas.

## Figures and Tables

**Figure 1 materials-16-02906-f001:**
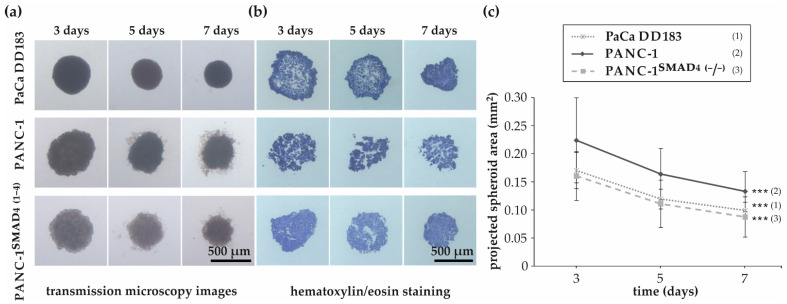
Generation of PDAC spheroids. Transmission microscopy images of spheroids (**a**) and cryosections of spheroids after hematoxylin/eosin staining (**b**) formed by PaCa DD183, PANC-1 and PANC-1^SMAD4 (−/−)^ cells after three, five and seven days of incubation. (**c**) Growth progression of spheroids between day 3 and 7. Data are expressed as the mean with standard deviation (*n* = 4–5, with nine replicates each). Statistical significances between spheroid sizes at day 3 and day 7 are indicated with ***. The respective confidential intervals are *p* ≤ 0.00005 and were calculated via the Student’s *t*-test. Abbreviations: (1), PaCa DD183; (2), PANC-1; (3), PANC-1^SMAD4 (−/−)^.

**Figure 2 materials-16-02906-f002:**
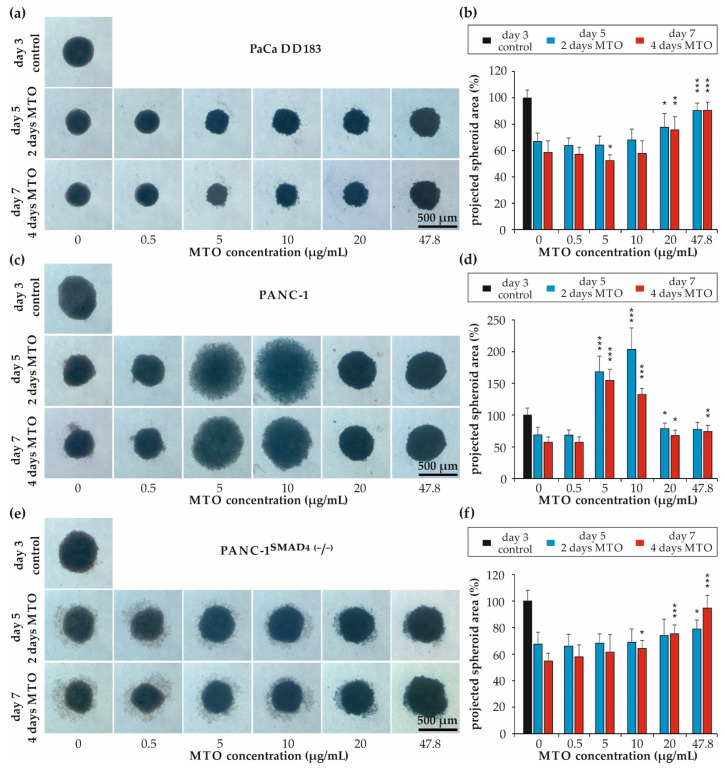
Effect of free MTO on pancreatic spheroids. Transmission microscopy images and quantification of representative spheroids formed by (**a**,**b**) PaCa DD183, (**c**,**d**) PANC-1 and (**e**,**f**) PANC-1^SMAD4 (−/−)^ spheroid cells after three, five and seven days of incubation. On day three, cell culture medium was supplemented with MTO (0, 0.5, 5, 10, 20 and 47.8 µg/mL, respectively). Growth progression was normalized to the spheroid size at day 3. Data are expressed as the mean with standard deviation (*n* = 4, with three replicates each). Statistical significances between MTO-free and MTO-treated spheroids are indicated with *, ** and ***. The respective confidential intervals are * *p* ≤ 0.05, ** *p* ≤ 0.0005 and *** *p* ≤ 3 × 10^−9^, respectively, and were calculated via the Student’s *t*-test. Abbreviations: MTO, mitoxantrone.

**Figure 3 materials-16-02906-f003:**
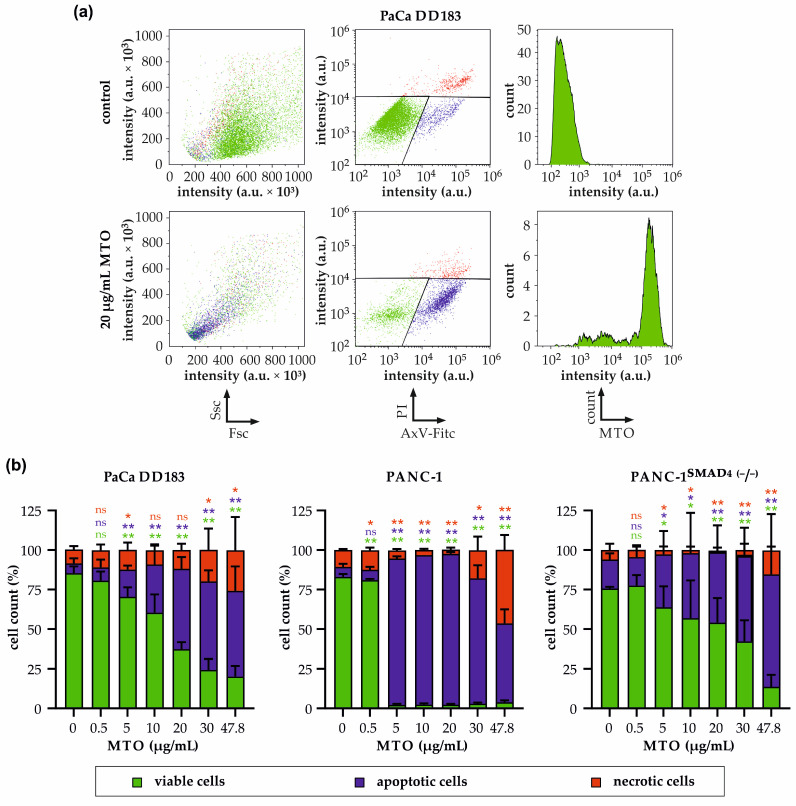
Influence of free MTO on PDAC tumor cell spheroids measured by flow cytometry. (**a**) Gating strategy. Shown are the representative plots and histograms of cells from PaCa DD183 spheroids after 4 days of treatment with 20 µg/mL MTO (second row) or with the corresponding amount of H_2_O (first row). Left panels exemplarily show cell morphology (forward scatter, Fsc, versus side scatter, Ssc), middle panels depict AnnexinV (AxV)-Fitc and propidium iodie (PI) staining, and right histograms represent mitoxantrone (MTO) uptake. (**b**) Cell viability of PaCa DD183, PANC-1, and PANC-1^SMAD4 (−/−)^ spheroids was determined by Annexin V-FITC/propidium iodide (AxV/PI) staining and analyzed by flow cytometry. Ax−/PI, Ax+/PI and PI+ cells were considered viable, apoptotic and necrotic, respectively. MTO concentrations were used as indicated. Statistical significances are indicated with * and **. The respective confidential intervals are * *p* ≤ 0.05 and ** *p* ≤ 0.0005, respectively, and were calculated via the Student’s *t*-test. Abbreviations: a.u., arbitrary unit; Ssc, side scatter, Fsc, forward scatter; count, detected events; MTO, mitoxantrone.

**Figure 4 materials-16-02906-f004:**
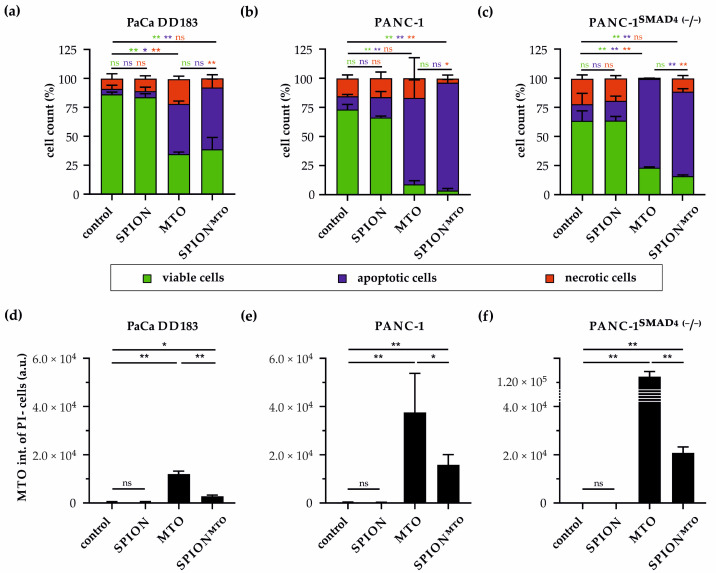
Effects of nanoparticle-bound MTO on different PDAC tumor cell spheroids. (**a**–**c**) Flow cytometric analysis of the viability of cells dissociated from (**a**) PaCa DD183, (**b**) PANC-1 and (**c**) PANC-1^SMAD4 (−/−)^ spheroids after treatment of H_2_O, SPIONs, free MTO or SPION-bound MTO was determined by Annexin V-FITC/propidium iodide (AxV/PI) staining. Necrotic, apoptotic and viable cells were determined by PI+, Ax+/PI and Ax−/PI cells, respectively. (**d**,**e**) MTO intensity of cells originating from (**d**) PaCa DD183, (**e**) PANC-1 and (**f**) PANC-1^SMAD4 (−/−)^ spheroids, measured by flow cytometry. Statistical significances between control, SPION, MTO and SPION^MTO^-treated spheroids are indicated with * and **. The confidential intervals are * *p* ≤ 0.05 and ** *p* ≤ 0.0005, respectively, and were calculated via the Student’s *t*-test. Abbreviations: SPIONs, superparamagnetic iron oxide nanoparticles; MTO, mitoxantrone; a.u., arbitrary unit; ns, not significan4.

**Table 1 materials-16-02906-t001:** Physicochemical characterization of superparamagnetic iron oxide nanoparticles. The hydrodynamic diameter and zeta potential of SPION and SPION^MTO^ were determined in H_2_O. Abbreviations: SPION, superparamagnetic iron oxide nanoparticles; MTO, mitoxantrone; Z ave. Ø, mean diameter.

Parameters	SPION	SPION^MTO^
Iron concentration (mg/mL)	4.78 ± 0.02	4.54 ± 0.02
Z ave. Ø in H_2_O (nm)	65 ± 1.9	61 ± 1.5
Zeta potential in H_2_O at pH ≈ 6.5 (mV)	−21 ± 0.7	−18 ± 0.3

## Data Availability

The data are available on reasonable request from the corresponding author.

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
