# Peer review of "Mitoxantrone and Mitoxantrone-Loaded Iron Oxide Nanoparticles Induce Cell Death in Human Pancreatic Ductal Adenocarcinoma Cell Spheroids"

_materials, 2023, doi:10.3390/ma16072906_

Round 1
Reviewer 1 Report
The manuscript "Mitoxantrone and Mitoxantrone-functionalized Iron Oxide Nanoparticles Induce Cell Death in Human Pancreatic Ductal Adenocarcinoma Cell Spheroids" is well written, reports interesting findings and has a good research and experimental design and description. It can be published in Materials after checking two issues:
Concerns:
1) A positive control needs to be added to Figures 3 and 4.
2) Table 1. The Z potential of SPION has two minus signs. Please check
Author Response
The manuscript "Mitoxantrone and Mitoxantrone-functionalized Iron Oxide Nanoparticles Induce Cell Death in Human Pancreatic Ductal Adenocarcinoma Cell Spheroids" is well written, reports interesting findings and has a good research and experimental design and description. It can be published in Materials after checking two issues:
Concerns:
- A positive control needs to be added to Figures 3 and 4.
In cell cultures, always a certain percentage of cells are dying or dead, thus, these cells serve as internal control for cell death staining in flow cytometry (Fig. 4). In figure 3 only the morphology and size of the spheroids is described, thus we did not add a positive control.
- Table 1. The Z potential of SPION has two minus signs. Please check
Thank you very much for this important hint. We corrected it.
We thank the reviewer for his/her positive evaluation and the important questions/remarks.
Reviewer 2 Report
Mitoxantrone and Mitoxantrone-functionalized Iron Oxide Nanoparticles Induce Cell Death in Human Pancreatic Ductal Ad- 3 enocarcinoma Cell Spheroids is an interesting manuscript. The author represents the every steps clearly and the novelty sounds good. The experimental parts also described well with well findings. Hence, I recommend the manuscripts for publications in Materials journal after minor English correction

Author Response
Mitoxantrone and Mitoxantrone-functionalized Iron Oxide Nanoparticles Induce Cell Death in Human Pancreatic Ductal Adenocarcinoma Cell Spheroids is an interesting manuscript. The author represents the every steps clearly and the novelty sounds good. The experimental parts also described well with well findings. Hence, I recommend the manuscripts for publications in Materials journal after minor English correction
We thank the reviewer for the positive evaluation. We corrected the manuscript for typos and mistakes.
Reviewer 3 Report
Dinter et al. have established so-called 3D tumor spheroid models of PDAC using two PDAC cell lines (PANC-1 and PaCa DD183). They also applied nanoparticles (SPIONS) for the delivery of a chemotherapeutic drug (MTO).
The rationale behind the study should be clarified, as this seems to be two studies in one paper. Apparently, the first aim was to test the importance of SMAD4 for the growth of PDAC derived cells (PANC-1) in 3D configuration. If so, then why is this not highlighted in the title of the present paper, and why does the abstract focus only on the use of SPIONS for drug delivery and not on this first aim (that is, the role of SMAD4 for the establishment of tumor spheroids)? The second aim is to evaluate whether SPIONS could enhance the delivery and/or efficacy of a chemotherapeutic drug (MTO) in 3D spheroids. To this end, the authors use the same parental PANC-1 versus SMAD4-deficient PANC-1 tumor spheroids. Free drug alone is also tested.
The use of the other cell model (PaCa DD183) is never clarified, and these cells are poorly described (in Methods, and in Results). Overall, while this paper addresses SMAD4, there is no doubt that KRAS mutations are also important, and PANC-1 is known to harbor oncogenic KRAS mutations. But the authors need to clarify the KRAS status of the in-house cell line as well.
The model is not well described. Hence, in the abstract and elsewhere the authors refer to the "pathophysiologically relevant microenvironment" of 3D tumor spheroids, but what is the evidence that the 3D models in the current study have a "microenvironment" (is there any production of extracellular martrix, do the authors add stromal (stellate) cells, or immune cells to the 3D spheroids to mimic the "environment" of actual PDAC tumors, or are they referring only to the fact that the cells grow in 3D in agarose-coated wells?
The authors repeatedly use the word "promising" in the abstract and in the main text when referring to the 3D cell spheroids or to the use of SPIONS for drug delivery (potentially, magnetic drug targeting, which was not studied here). They should refrain from using this term over and over again and simply describe the model, and the results. Again, the authors need to reorganize the manuscript (title, abstract, main text) to make it more clear that they are testing 2 different hypotheses, or addressing 2 different objectives: First, does SMAD4 control the grwoth or development of 3D spheroids of the PANC-1 cell line and/or the response to chemotherapeutic treatment (MTO)? Second, can SPIONS be used to deliver MTO more effectively in 3D tumor models of PDAC (and does SMAD4 play a role)? As already mentioned, the authors need to claify why they choose to add one more PDAC cell line (PaCa DD183) (and its KRAS and SMAD4 status); is this cell line more or less agressive than PANC-1; more or less responsive (or resistant) to treatment? Have the authors tested a more conventional drug, e.g., gemcitabin to address possible differences between PANC-1 and PaCa?
The particles: The SPIONS may have been studied before, but some basic information is missing such as: the primary particle size (by TEM, not DLS). The authors mention in Methods that the "functionalization" of SPIONS with MTO has been previously described: how does MTO bind to the SPIONS, and is this truly "functionalization" (this suggests some sort of covalent binding)? Have the authors studied the drug loading and drug release properties of these SPIONS? How stable are the SPIONS in biologically relevant media? Have the authors studied the Fe-content in 3D spheroids with/without SMAD4 (e.g., by using ICP-MS of the dissociated tumour spheroids)? The authors suggest that "there is a higher uptake of free and particle bound drug into the SMAD4 deletion mutant cells". So the question is: do the SPIONS actually deliver the drug (MTO) into cells, or is the drug released from the nanoparticles and passively taken up by the cells (through a mechanism that is somewho SMAD4-dependent); do the nanoparticles make a difference? In fact, based on the results in the final section of the paper, it seems clear that all 3 tumor spheroid models responded similarly to free drug versus SPION-linked MTO suggesting that nanoparticles do not make a difference. The difference, which should have been highlighted in the title of the paper, is that SMAD4 appears to affect the susceptibility to drug treatment (irrespective of whether or nor the drug is "delivered" via nanoparticles). Indeed, the Discussion section seems to address only the utility of 3D tumor spheroids, and there is almost nothing on SPIONS (for drug delivery/magnetic drug targeting, or perhaps for imaging/theranostics).
Author Response
Reviewer 3:
Dinter et al. have established so-called 3D tumor spheroid models of PDAC using two PDAC cell lines (PANC-1 and PaCa DD183). They also applied nanoparticles (SPIONS) for the delivery of a chemotherapeutic drug (MTO).
The rationale behind the study should be clarified, as this seems to be two studies in one paper. Apparently, the first aim was to test the importance of SMAD4 for the growth of PDAC derived cells (PANC-1) in 3D configuration. If so, then why is this not highlighted in the title of the present paper, and why does the abstract focus only on the use of SPIONS for drug delivery and not on this first aim (that is, the role of SMAD4 for the establishment of tumor spheroids)? The second aim is to evaluate whether SPIONS could enhance the delivery and/or efficacy of a chemotherapeutic drug (MTO) in 3D spheroids. To this end, the authors use the same parental PANC-1 versus SMAD4-deficient PANC-1 tumor spheroids. Free drug alone is also tested.
Actually, the rationale behind the study was to investigate the role of SPIONs for drug delivery. But the reviewer is correct, the focus should be set more clearly. We tried to focus on the delivery of the drug by magnetic nanoparticles, whereas the cell lines only serve as tools for investigation. Thus, we shifted the initial figure 1 in the Appendix.
The use of the other cell model (PaCa DD183) is never clarified, and these cells are poorly described (in Methods, and in Results). Overall, while this paper addresses SMAD4, there is no doubt that KRAS mutations are also important, and PANC-1 is known to harbor oncogenic KRAS mutations. But the authors need to clarify the KRAS status of the in-house cell line as well.
Thank you for this important comment. The cell line was chosen to analyse possible effects on primary cell lines compared to established cell lines. However, these cells are extremely difficult to cultivate and we are currently analysing the KRAS and the SMAD4 status of these cells by different methods. The Panc-1 were included since we were able to generate SMAD4 knock out clones. It is well established that SMAD4 deletion is associated with shorter survival of patients and might play a role in metastasis therefore, we were interested how such cells behave in this assay.
The model is not well described. Hence, in the abstract and elsewhere the authors refer to the "pathophysiologically relevant microenvironment" of 3D tumor spheroids, but what is the evidence that the 3D models in the current study have a "microenvironment" (is there any production of extracellular martrix, do the authors add stromal (stellate) cells, or immune cells to the 3D spheroids to mimic the "environment" of actual PDAC tumors, or are they referring only to the fact that the cells grow in 3D in agarose-coated wells?
The spheroids consist only of the respective pancreatic tumor cells. The author is absolutely correct, in the microenvironment of in vivo tumors are several additional factors, which should be considered as well. These additional factors like cancer associated fibroblasts and immune cells should be added to our spheroids in future studies to improve the experimental setting.
In the revised manuscript, we better described our current model and highlighted the advantages over two-dimensional cell cultures, even if they are composed only of tumor cells. We also addressed the limitations of our current spheroid model in the discussion.
The authors repeatedly use the word "promising" in the abstract and in the main text when referring to the 3D cell spheroids or to the use of SPIONS for drug delivery (potentially, magnetic drug targeting, which was not studied here). They should refrain from using this term over and over again and simply describe the model, and the results.
We thank the reviewer for the helpful comment. We reduced the use of the term „promising“ to establish a more desciptive language in our manuscript.
Again, the authors need to reorganize the manuscript (title, abstract, main text) to make it more clear that they are testing 2 different hypotheses, or addressing 2 different objectives:
First, does SMAD4 control the grwoth or development of 3D spheroids of the PANC-1 cell line and/or the response to chemotherapeutic treatment (MTO)? Second, can SPIONS be used to deliver MTO more effectively in 3D tumor models of PDAC (and does SMAD4 play a role)?
As already mentioned, the authors need to claify why they choose to add one more PDAC cell line (PaCa DD183) (and its KRAS and SMAD4 status); is this cell line more or less agressive than PANC-1; more or less responsive (or resistant) to treatment? Have the authors tested a more conventional drug, e.g., gemcitabin to address possible differences between PANC-1 and PaCa?
As explained previously, our aim was to test if MTO-loaded SPIONs can effectivley induce cell death in pancreas carcinoma spheroids. The different cell lines only served as tools for investigation. Thus, we have tried to shift the focus away from the cell lines to the magnetic drug targeting.
We agree with the reviewer, that mitoxantrone is a not a standard drug in treatment of pancreas carcinomas. There several reasons for using mitoxantrone in the study. At first it is very effective and therefore often inducing systemic side effects in the patients. We hypothesize that by applying it with magnetic nanoparticles one can exploit the effectivity of the drug and simultaneously reduce the side effects. Beyond that, mitoxantrone is a chemically related to doxorubicin, a drug which is used very often in combination with nanoparticles for cancer treatment. In fact, there is currently a running clinical study using liposomal encapsulated doxorubicin for treating pancreatic cancer led by the University of Oxford (PanDox: Targeted Doxorubicin in Pancreatic Tumours (PanDox); ClinicalTrials.gov Identifier: NCT04852367). Other than doxorubicin, which is very chemically very unstable, mitoxantrone is quite stable and by that very well to be analyzed even in complex media or extracted from tissue (Tietze et al. Journal of Biomedicine and Biotechnology, 2010: Mitoxantrone loaded superparamagnetic nanoparticles for drug targeting: a versatile and sensitive method for quantification of drug enrichment in rabbit tissues using HPLC-UV)
Furthermore, MTO is known as inductor of immunogenic cell death, inducing the release of DAMPs (damage-associated molecular patterns) from the dying cells, which lead to activation of immune cells such as dendritic cells. We previously showed that the loading of MTO onto SPIONs does not influence the capability of the drug to induce immunogenic cell death in tumor cells, leading to subsequent activation of dendritic cells (Ratschker 2020 Pharmaceutics: Mitoxantrone-Loaded Nanoparticles for Magnetically Controlled Tumor Therapy–Induction of Tumor Cell Death, Release of Danger Signals and Activation of Immune Cells, Alev 2018, Journal of Controlled Release: Targeting of drug-loaded nanoparticles to tumor sites increases cell death and release of danger signals). When the chemotherapeutic drug MTO can be accumulated in the tumor region, the immune system can be spared from the toxic effects of the drug (Janko Int J Mol. Sci 2013: Magnetic Drug Targeting Reduces the Chemotherapeutic Burden on Circulating Leukocytes). When then immunogenic cell death is induced in the tumor region, the preserved immune cells can induce a long-term anti-tumor immune response.
The engagement of the immune system by use of an immunogenic cell death inductor conjugated on nanocarriers for the treatment of pancreas cancer has been shown by others before (Lu et al Nature Communications 2017, Nano-enabled pancreas cancer immunotherapy using immunogenic cell death and reversing immunosuppression).
The particles: The SPIONS may have been studied before, but some basic information is missing such as: the primary particle size (by TEM, not DLS). The authors mention in Methods that the "functionalization" of SPIONS with MTO has been previously described: how does MTO bind to the SPIONS, and is this truly "functionalization" (this suggests some sort of covalent binding)? Have the authors studied the drug loading and drug release properties of these SPIONS?
Mitoxantrone is not covalently bound to the SPIONs, it is bound by adhesion. Thus, we replaced the term „functionalization“ by „loading“. We added further information on the particle features such as primary particles size determined by TEM. Previously, drug loading and release properties of the SPIONs have been determined. We also added this information in the revised version of the manuscript (see 3.1. “Physicochemical SPION characterization”).
How stable are the SPIONS in biologically relevant media?
We previously investigated the colloidal stability of the SPIONs in biologically relevant medium such as serum-containing media and whole blood. We found that the albumin protein corona strongly increases the stability of the particles, so that no agglomeration was detected, even in the presence of magnetic fields (Bilyy 2018, Frontiers in Immunology: Inert Coats of Magnetic Nanoparticles Prevent Formation of Occlusive Intravascular Co-aggregates With Neutrophil Extracellular Traps). We added this information in the revised version of the manuscript (see 3.1. “Physicochemical SPION characterization”).
Have the authors studied the Fe-content in 3D spheroids with/without SMAD4 (e.g., by using ICP-MS of the dissociated tumour spheroids)? The authors suggest that "there is a higher uptake of free and particle bound drug into the SMAD4 deletion mutant cells". So the question is: do the SPIONS actually deliver the drug (MTO) into cells, or is the drug released from the nanoparticles and passively taken up by the cells (through a mechanism that is somewho SMAD4-dependent); do the nanoparticles make a difference? In fact, based on the results in the final section of the paper, it seems clear that all 3 tumor spheroid models responded similarly to free drug versus SPION-linked MTO suggesting that nanoparticles do not make a difference. The difference, which should have been highlighted in the title of the paper, is that SMAD4 appears to affect the susceptibility to drug treatment (irrespective of whether or nor the drug is "delivered" via nanoparticles).
We did not measure the Fe-content in the spheroids. From our flow cytometry data we cannot discriminate if the MTO taken up by the cells is in free or nanoparticle-bound form.
In previous investigations using spheroids from HT-29 colon carcinoma cells using fluorescence microscopy we observed a delayed penetration of MTO into the tumor spheroids when applied as SPION-MTO (Hornung et. al. 2015, Molecules: Treatment Efficiency of Free and Nanoparticle-Loaded
Mitoxantrone for Magnetic Drug Targeting in Multicellular Tumor Spheroids). This was a general phenomenon in our investigations: when MTO was applied as SPION-MTO, the observed effects such as uptake, DNA degradation, cell death induction were slightly delayed.
So far we do not know if MTO is released from the particles or if the particles penetrate into the spheroid together with the chemotherapeutic cargo. At least in the beginning, the MTO must have been bound, otherwise we would not have seen differences in uptake velocity and amount. We addressed this question in the discussion. We also do not know, which role SMAD4 plays for MTO and particle uptake. This is subject of future investigations.
Indeed, the Discussion section seems to address only the utility of 3D tumor spheroids, and there is almost nothing on SPIONS (for drug delivery/magnetic drug targeting, or perhaps for imaging/theranostics).
We addressed the use of SPIONs for biomedical applications in the discussion.

Reviewer 4 Report
The work investigates the potential of using superparamagnetic iron oxide nanoparticles (SPIONs) as a drug delivery system for mitoxantrone (MTO) using 3D culture of human pancreatic ductal adenocarcinoma cell lines. The theme discussed is very interesting and important and the study performed presents relevant data.
The manuscript is well written and organized. I strongly recommend it for publication.
However, there a few mistakes that must be corrected, such as:
- The volume unit millilitres is represented sometimes by ml and other by mL. Please correct in the text as well as in figures and in figure legends to be consistent.
- In section 2.1 Materials, the reagent lauric acid is not mentioned.
- In line 147 is written “…was vortexed with 100 μL mL mitoxantrone solution…” please delete mL.
- In line 148 is stated that the obtained concentration of iron is 4.84 mg/mL. The final iron concentration cannot be higher than the concentration of the initial iron solution that is stated as 4.78 mg/mL. Please correct the iron concentration of the final or the initial solution.
- In table 1, the zeta potential of SPION is written as “- - 21”.
- Line 297: it is written SMDAD4. Please correct.
- Line 344: Please correct the concentration units of MTO to g/mL
- Line 368: It is written “… Ax+PI+ are considered viable, Ax+PI- apoptotic and PI+ cells necrotic”. Correct the Ax+PI+ to Ax-PI-
- References: Correct the references in accordance with the rules of the journal. Materials.
Author Response
Reviewer 4:
The work investigates the potential of using superparamagnetic iron oxide nanoparticles (SPIONs) as a drug delivery system for mitoxantrone (MTO) using 3D culture of human pancreatic ductal adenocarcinoma cell lines. The theme discussed is very interesting and important and the study performed presents relevant data.
The manuscript is well written and organized. I strongly recommend it for publication.
Thank you for the positive evaluation.
However, there a few mistakes that must be corrected, such as:
- The volume unit millilitres is represented sometimes by ml and other by mL. Please correct in the text as well as in figures and in figure legends to be consistent.
We corrected ml/mL for consistency.
- In section 2.1 Materials, the reagent lauric acid is not mentioned.
We added lauric acid.
- In line 147 is written “…was vortexed with 100 μL mL mitoxantrone solution…” please delete mL.
We deleted ml.
- In line 148 is stated that the obtained concentration of iron is 4.84 mg/mL. The final iron concentration cannot be higher than the concentration of the initial iron solution that is stated as 4.78 mg/mL. Please correct the iron concentration of the final or the initial solution.
We apologize for the mistake. We corrected the initial (4,84 mg/ml) and final (4,36 mg/ml) iron concentration.
- In table 1, the zeta potential of SPION is written as “- - 21”.
We corrected it.
- Line 297: it is written SMDAD4. Please correct.
We corrected it.
- Line 344: Please correct the concentration units of MTO to mg/mL
We corrected it.
- Line 368: It is written “… Ax+PI+ are considered viable, Ax+PI- apoptotic and PI+ cells necrotic”. Correct the Ax+PI+ to Ax-PI-
We corrected it.
- References: Correct the references in accordance with the rules of the journal. Materials.
We adapted the formation of the references according to the journal style.
We thank the reviewer for his/her positive evaluation and the important questions/remarks.